# Urinary Tract Infection in Children: An Up-To-Date Study

**DOI:** 10.3390/biomedicines12112582

**Published:** 2024-11-12

**Authors:** Silvio Maringhini, Demet Alaygut, Ciro Corrado

**Affiliations:** 1Department of Pediatrics, ISMETT (Istituto Mediterraneo per i Trapianti e Terapie ad Alta Specializzazione), Via Ernesto Tricomi, 5, 90127 Palermo, Italy; 2Department of Pediatric Nephrology, Izmir Faculty of Medicine, University of Health Sciences, Gaziler Street No 1, Izmir 35180, Turkey; 3Pediatric Nephrology Unit, Ospedale “G. Di Cristina”, ARNAS “Civico” UOC, Piazza Porta Montalto 2, 90127 Palermo, Italy

**Keywords:** urinary tract infection (UTI), congenital abnormalities of the kidney and urinary tract (CAKUT), chronic kidney disease (CKD) in children

## Abstract

Urinary tract infections (UTIs) are common bacterial infections in children. UTIs may be limited to the bladder or involve the kidneys with possible irreversible damage. Congenital abnormalities of the kidney and urinary tract (CAKUT) are often associated with UTIs; kidney scars have been considered a consequence of untreated UTIs but may be congenital. The mechanism by which bacteria produce inflammation in the urinary system has been intensively investigated. Diagnostic tools, including invasive imaging procedures, have been advocated in infants and small children with UTIs but are not necessary in most cases. Effective antibiotic drugs are available, and prophylactic treatment has been questioned. Several guidelines on UTIs are available, but a simple one for general practitioners is needed.

## 1. Introduction

Urinary tract infections (UTIs) are the second most common bacterial infections in children. Symptoms of UTIs are present in older children but are less evident in newborns, infants, and small children. In some cases, bacteriuria may be isolated; in others, microbes invade the bladder and the kidneys. The mechanism by which bacteria produce inflammation in the urinary system and its defense against bacteria are not well known. Congenital abnormalities of the kidney and urinary tract (CAKUT) are often associated with UTIs, and permanent kidney damage may be the consequence of UTIs. In recent decades, many reviews have analyzed the utility of diagnostic tests, imaging procedures, and the results of treatment trials dealing with UTIs in children, meaning that more information is now available. Several guidelines on UTIs are available, but many problems have not been solved. In this review, we will briefly report on our knowledge of UTIs and the questions that still remain open. Our review is not a systematic one; we performed a PubMed search of original papers and reviews dealing with the various aspects of UTIs, concentrating on the most controversial topics that need further studies. We also reviewed the most recent guidelines with the aim of stimulating the production of a European consensus.

## 2. Epidemiology and Definitions 

In the first decade of life, almost 8% of girls and 2% of boys have a UTI. Roughly 30% of children with UTIs will have a second infection [1]. The urine produced by the kidney is sterile; bacteria may reach the kidneys and the urinary tract rarely via the bloodstream or, more commonly, via the retrograde ascent from the perineal area to the bladder and eventually to the kidneys in the presence of an abnormal ureteric junction. In some cases, bacteria proliferate but do not produce an inflammatory response (asymptomatic bacteriuria), while in most cases, they trigger the immune system and produce a UTI. The urinary bladder hosts a variety of microbes that constitute the urobiome, as culture-dependent and sequencing methodologies can reveal [2,3]; its role may be relevant in protecting from infections.

The bacterial aggression of the urinary tract produces symptoms (dysuria, voiding urgency and frequency, hematuria, lumbar pain, and fever) in older children, while signs and symptoms of UTIs in infants are nonspecific: lethargy, irritability, anorexia, fever, and neonates may develop bacteremia [4]. UTIs should be suspected in all infants with a fever of unknown origin; on the other hand, fever may be absent in a very sick neonate. The incidence of UTIs is greater in uncircumcised males during the first year of life; on the other hand, a UTI may provide an early discovery of a CAKUT in uncircumcised children. An infection restricted to the bladder is called cystitis, which is an infection of the upper urinary tract, and an infection of kidney is called pyelonephritis. *Escherichia coli* is the most common pathogen in a UTI, but other Gram-negative (namely *Klebsiella pneumoniae*, *Pseudomonas aeruginosa*, *Proteus mirabilis*, *Enterococcus faecalis*, *Citrobacter*) and Gram-positive bacteria (*Staphylococcus coagulase-negative*, *Enterococcus*, *Staphylococcus aureus*) can also cause pyelonephritis. An infection in the kidney may produce irreversible damage, and a prompt diagnosis is desirable. A fever (body temperature higher than 38 Celsius degree) is the most useful clinical sign of pyelonephritis. Interstitial cystitis/bladder pain syndrome (IC/BPS) is a rare disease characterized by avoiding symptoms (bladder pain, urgency, frequency, nocturia) and may affect children and adolescents; its diagnosis and management are challenging [5]. 

## 3. Kidney Involvement in UTI

The mechanisms by which infections develop, progress, or are eliminated in the kidney are not completely known. Virulence factors produced by uropathogenic *Escherichia coli* (UPEC) are known, but not those for other bacteria. UPEC strains produce P fimbriae, which attach to kidney epithelial cells, but also α-haemolysin, F1C/S fimbriae, and aerobactin. Kidney epithelial cells, in response to a UPEC infection, activate receptors that stimulate the production of inflammatory chemokines and cytokines by neutrophils and monocytes and activate innate immune signaling and the complement system with the production of antimicrobial peptides (AMPs) and antimicrobial proteins (defensins, cathelicidin, metal-binding proteins, ribonucleases, and uromodulin) [6]. Recent studies have shown how cellular immune responses inhibit bacterial attachment and lyse-invading pathogens [7,8]. RNase, a macrophage-derived protein, has antimicrobial activity toward UPEC. RNase6 is produced by a heterogeneous gene, which has a protective role, as shown in experimental UTI [9]; it has been found that the R66 variant, encoded by the major SNP rs1045922 allele, has a vigorous bactericidal activity [10]. A UPEC infection induces a bladder epithelial cell line to secrete exosomes, which are captured by macrophages that produce proinflammatory cytokines [11]. Lectin collectin 11 (CL-11), produced by the kidney, has an important role in protecting against kidney and bladder infections and may have a therapeutic role [12]. On the other hand, immune cells may produce kidney injury. Immunomodulation therapy to contrast “excessive inflammation” may be an alternative to antimicrobial treatments.

### 3.1. CAKUT

CAKUTs are often associated with UTIs [13]. A CAKUT may promote a UTI by inadequate urine flow, incomplete emptying of the bladder, and a reflux of contaminated urine. The most common CAKUT is vesico-ureteral reflux (VUR), which is detected in about one-third of children with a UTI. VUR may be primitive due to incompetent anatomic junctions and generally improves with time. Secondary RVU may be caused by surgical procedures such as a renal transplant, the surgical correction of a vesicoureteral obstruction, and other bladder anatomic or functional abnormalities. A strong association between bladder/bowel dysfunction (BBD) and recurrent UTIs has been described in preschool children, and an assessment of BBD is recommended in small children with pyelonephritis [14]. Transient VUR may occur as a consequence of increased pressure in the bladder in patients with BBD or in the acute phase of a UTI.

### 3.2. Kidney Scarring 

It has been proven that UTIs can cause kidney scarring. Renal scars are present in 15% of children with VUR, and this condition is referred to as reflux nephropathy (RN). Renal scarring associated with VUR may be congenital or a result of acute pyelonephritis. It is difficult to differentiate a congenital RN from an acquired RN: congenital RN has large scars and is more common in boys; the acquired RN has smaller scars, usually in the upper poles of the kidney, and is more common in girls [15]. The mechanism leading to scarring in the kidneys is poorly understood and is likely multifactorial. It was hypothesized that scarring in the kidneys occurs as a result of localized infection, causing the release of inflammatory mediators like cytokines chemokines, bacterial toxins/peptides, and complement proteins. An overactive innate immune response in UTIs may produce irreversible kidney damage. The mechanisms of kidney fibrosis are complicated, and genetic and epigenetic mechanisms regulate kidney fibrosis. Activated myofibroblasts are considered to be the major producers of fibrosis. Experimental evidence shows that dysregulated macrophage-dependent inflammatory responses promote kidney damage and fibrosis [16]. Several risk factors for renal scarring in children affected by UTIs have been proposed: bacteria virulence factors, intra-renal reflux, recurrent febrile UTIs, delays in therapy, a high VUR grade, and bladder and bowel dysfunctions (BBDs). The fact that a renal scar is a rare occurrence in adults with UTIs suggests that a genetic predisposition is involved in kidney damage. Children aged two years and older have significantly greater odds of kidney scarring compared to those of a younger age. The intra-calyceal pressure increases with age; in the presence of a VUR, papillary damage may be produced in small children with intra-renal reflux but also in the absence of a UTI. The development of a new scar in young adults has not been proven, with the exception of transplanted patients [17,18]. Recurrent episodes of pyelonephritis may produce renal scars, as shown in collaborative studies [19,20]. A delay in the assumption of antibiotic treatment has been considered a cause of permanent kidney damage, but it is not clear how many days of infection are needed to produce a renal scar [21]. Children with a VUR higher than grade III are likely to develop scarring compared to children with lower grades [22]. A large number of children with VUR (from 25% to 68%) have bladder detrusor overactivity and/or sphincter overactivity; both are associated with a consistent increase in bladder pressure, which may be the cause of reflux [23]. The role of uromodulin, the most represented protein in human urine, in contrast to bacterial proliferation and kidney damage, deserves more study; in a recent paper, children with VUR carrying UMOD genotype rs4293393 TC were more prone to developing renal scars [24]. The development of abnormal gut microbiota during infancy may increase the risk of febrile UTI [25,26]. 

### 3.3. Late Sequelae of UTI 

Old studies showed a high incidence of hypertension and a decrease in renal function in patients with a UTI followed for 5 to 20 years. These complications occurred in patients who already had renal damage at the onset of a UTI. One study compared 53 children with scars to 51 without after their first UTI and did not notice any difference in blood pressure between the two groups [27]. In 72 women who had a history of UTIs in their childhood, hypertension was diagnosed in 10 and preeclampsia in 4 out of 151 pregnancies in those who had kidney damage before a UTI [28]. BBD was detected in 37% of 73 children with a UTI and VUR at 3-4 years of age, decreasing to 23% in adolescence [29]. Chronic kidney disease has been considered a consequence of UTIs associated with VUR for a long time, but it seems that it may happen only in children who already have kidney damage [17,30]. In conclusion, preventing kidney damage in children with UTIs is important; several factors have been documented as responsible for kidney scarring, but further studies should concentrate on individual predisposition. 

## 4. Diagnosis

### 4.1. Laboratory

Urine analysis is the main procedure diagnose a UTI. Some problems are related to proper urine collection and bacterial count. It is recommended to obtain a urine sample by either bladder catheterization (BC) or suprapubic aspiration (SPA) in infants and small children. Specimen contamination is more likely with BC; however, bladder catheterization is usually preferred because SPA is more invasive. For toilet-trained children, a midstream clean catch urine specimen may be collected [31]. Collecting urine using a bag is simple and non-invasive but bag samples have a high rate of contamination and should not be used for culture. A bag specimen may be used for an initial analysis, and if it is abnormal, a new specimen can be obtained by catheterization or SPA [32,33] (Table 1). Urine dipsticks are easy to obtain, and dipstick analysis includes testing for leukocyte esterase (a marker for pyuria) and nitrite (a marker for Enterobacteriaceae), which is helpful, but the final assessment should be performed with a urine culture [34]. The urinary neutrophil gelatinase-associated lipocalin (NGAL) test has a higher sensitivity, but it has an increased cost; further studies are needed to determine the utility of urinary NGAL as a screening test for UTI [35]. The number of colonies is fundamental for the diagnosis of UTI and can be obtained by a urine culture. A UTI is defined as the growth of a single pathogen with a colony count of ≥50,000 CFU/mL or a colony count between 10,000 and 50,000 CFU/mL, with associated pyuria detected on urinalysis made on urine samples obtained via bladder catheterization in older children while the optimal definition for a UTI in neonates has not been established. In the case of SPA, any growth colony count of 1000 CFU/mL of a urinary pathogen is significant. Future studies are needed to determine the most convenient method of urine sampling and colony counts to diagnose a UTI [36] (Table 2). The culture of urine samples for bacterial or fungal infections requires a long time (48 h), and broad-spectrum antibiotics are often prescribed in children with a positive dipstick, which may produce antimicrobial resistance. Urinary volatile organic compounds (VOCs) may serve as potential biomarkers for diagnosing UTIs. Mass spectrometry and the electronic nose (eNose) are techniques used for the chemical analysis of volatile compounds; eNoses are less specific and less precise compared to mass spectrometry but less expensive and allow for a quick analysis of urinary tract infections as a bedside test. The distinction between non-infected and infected urine may be accurately detected, but the type of infection cannot always be distinguished. Taking into account its limitations, we believe that eNose has the potential to become a feasible device to be applied in clinical practice for the diagnosis of UTIs, in particular in identifying patients with an abnormal dipstick who do not have a UTI [37]. 

Several blood and urinary biomarkers have been associated with kidney fibrosis. Specific biomarkers can predict the development of renal scars. Pentraxin 3 (PTX3) is a mediator of inflammation. The urinary PTX3 concentration is elevated in patients with renal scars [47]. NGAL is produced in increased amounts and excreted in the urine in cases of renal tubular damage; it is an indicator of renal scar formation in patients with a VUR [48]. A multiplex polymerase chain reaction (M-PCR) has increased sensitivity for microbial detection in UTIs. In a study, M-PCR was shown to be useful when infection was caused by organisms other than Escherichia coli [49]. 

In the past, many studies have been conducted in an attempt to distinguish between upper and lower UTIs. Procalcitonin (PCT) represents the most reliable blood test for the diagnosis of pyelonephritis [50]. Viral infections, however, can reduce the sensitivity of PCT [51]. A number of urinary markers have been shown to increase during pyelonephritis as opposed to cystitis (INF gamma, IL15, and chemokine ligands). Experimental data have shown how to distinguish acute pyelonephritis (APN) from acute cystitis at the molecular level [52]. Fever (body temperature higher than 38 degrees Celsius) is the most useful clinical sign to differentiate APN from cystitis. In any case, a prompt and adequate treatment of a febrile UTI may avoid the need to differentiate between an upper and lower UTI. Blood culture collection in pediatric patients with UTI is of limited utility [53]. 

### 4.2. Imaging

In the past, the majority of children with a UTI were submitted to radiological investigations (including ultrasound, voiding cystogram, scintigraphy, urography, computerized tomography, or magnetic resonance) in order to detect urinary malformations, which may produce permanent kidney damage. It has been proven that a kidney scar may be a consequence of a UTI, but in most cases, the parenchymal damage is congenital or due to severe obstruction. 

The criteria adopted for performing an ultrasound of the urinary apparatus (KUS) following a febrile UTI are different; some guidelines propose a KUS in children under 2 years of age; other guidelines restrict KUS to <6 months of age or do not recommend KUS except in the presence of other risk factors such as a UTI caused by abnormal pathogens or recurrent UTIs [39,41,54,55,56,57]. 

A voiding micturating urethra-cystogram (VCUG) allows VUR, urethral valves, or bladder abnormalities to be revealed. VCGU is usually performed in small children following a febrile UTI if the US is abnormal or the infections are recurrent [41].

Renal imaging with a radionucleotide dimercaptosuccinic acid (DMSA) scan can be used to confirm pyelonephritis or to detect renal scarring; in the first case, the scan should be performed within 1 week following a febrile UTI. In the second case, it should be undertaken 6–12 months later. The DMSA may help to diagnose congenital renal hypo dysplasia, which has a different appearance from that of acquired scarring. Guidelines on UTIs differ in their recommendation to perform a DMSA scan; some do not advise DMSA, while others do in case of an atypical infection or high-grade VUR [39]. A retrospective study suggests that deep learning (DL) has the potential to differentiate normal from abnormal kidneys in children using 99mTc-DMSA SPECT imaging [58] (Table 3) 

Computed tomography (CT) provides a good resolution of the urinary tract. Sedation is often not necessary, but an IV contrast is needed in most cases. Radiation exposure is an important risk of CT, and its use should be limited to cases where the study is important for a certain diagnosis. The utility of non-contrast CT for the diagnosis of acute pyelonephritis has been evaluated recently in a retrospective study in older patients; a prospective study might be performed in children [59]. 

Magnetic resonance studies (MRU) are not usually recommended in children with UTIs due to the need to sedate the child. In the future, MRUs may become easier to perform and cost-effective [60]. MRU may be a more suitable investigation in some cases, given the lack of radiation exposure [61]. 

Urodynamic studies (UDSs) allow bladder dysfunctions, which have an important role in producing UTIs and renal scars in children with a VUR, to be identified [62]. UDSs require the use of catheters; catheter-free UDSs would be useful, and ambulatory wireless devices that detect pressure or volume might be a useful tool in the future [63]. Voiding charts and urinary flux metric studies on bowel habits should be conducted in children with recurrent UTIs. 

## 5. Treatment

### 5.1. Antibiotics

The discomfort of a UTI may be relieved by antibiotic treatment. A large number of antibiotics are effective in treating UTIs. The type of drug is usually guided by a urine culture, but most antibiotics are effective. It has been proven that oral administration is effective unless the patient is not taking the drug. The length of treatment should be 3–5 days in case of cystitis and 10–14 days in case of pyelonephritis, limiting intravenous antibiotics to a few days followed by oral therapy [64,65]. In young infants with UTIs, a short course of parenteral antibiotics is as safe as long courses [66]. A recent randomized, controlled trial has shown that a 5-day course is comparable with a 10-day course of oral amoxicillin–clavulanate in children with a febrile UTI [67], although another trial showed that children assigned to a 5-day course of therapy were more likely to have asymptomatic bacteriuria or a positive urine culture than those of a longer course at follow up [68] (Table 4).

### 5.2. Chemoprophylaxis 

The recurrence of UTIs in a considerable percentage of cases has led to the fear that renal damage may ensue and prophylactic antibiotic treatment (CAP), has been suggested, particularly in children with VUR. Such a policy has been largely used in the past and criticized afterward but recent trials show that chemoprophylaxis is effective at reducing the number of UTIs, but not renal damage, at least over a short period of observation [69]. The cost, side effects, and the occurrence of antimicrobial resistance are possible risks of CAP, suggesting its limitations for use in children with major obstructive uropathies before surgery or according to each patient’s specific risk factors [70,71,72]. An international multicenter randomized controlled trial (PREDICT) has shown that in infants with grade III, IV, or V vesicoureteral reflux and no previous UTIs, CAP was effective in preventing UTIs despite an increased occurrence of non-*E. coli* infections and antibiotic resistance [73] (Table 4). 

### 5.3. Immunomodulators

Immunomodulatory treatment involving immune response regulators has been developed and can be an alternative to antibiotics [74]. The administration of corticosteroids, in addition to antimicrobial therapy, seems to reduce kidney scar formation in children with acute pyelonephritis [75], but NSAIDs might be associated with worse outcomes. Some studies have shown that Vitamin A or E supplements, such as L-carnitine, may reduce scarring in kidneys [76].

Cranberry products may reduce the risk of UTIs in children, but there is no evidence that Cranberry products are effective for the treatment of UTIs [77,78].

Disappointing results have been described with drugs acting on bladder dysfunctions, while a recent study indicates that botulinum toxin-A may be useful for therapy-refractory children with dysfunctional voiding [62,79]. Promising results have been obtained for randomized control trials with cystitis and pyelonephritis prevention using vaccines developed against UPEC [80,81,82]. Beyond vaccines, minimizing UPEC attachment is a promising strategy for preventing UTIs. The prophylactic administration of D-mannose reduces recurrent UTIs. 

Probiotics have direct bacterial inhibitory activities (e.g., secretion of antimicrobial substances, including the production of an acidic environment and competition for nutrients) and indirect bacterial inhibitory actions (e.g., immunomodulatory activation of host immune cells) and can limit the growth of antibiotic-resistant bacteria; as an example, Lactobacillus spp., a natural microbiota of the urinary tract, may interfere with the pathogenicity of P. mirabilis [83]. The utility of probiotics such as anti-UPEC in the prevention of UTI has been recently reviewed [84]. Bacteriophages are used to introduce genetic material into bacteria and can produce bacterial lysis [85]. Among the recently developed antibiotics, Gepotidacin has bactericidal activity capable of targeting DNA and shows efficacy in treating uncomplicated UTIs [86]. Finally, many Chinese herbal formulas have been shown to reduce kidney fibrosis and renal scarring in UTIs [87]. Asymptomatic bacteriuria should not be treated except in special circumstances [88]. 

A large effort should be made to reduce the use of antibiotics in children with UTIs. Costs, side effects, and antibiotic resistance are the main reasons to limit antibiotic use in children with febrile UTIs. Antibiotic prophylaxis should be limited to subjects who are at risk of recurrent UTIs and kidney scars; new genetic studies might help those children. Immunomodulators and probiotics may be an alternative when prospective case–controlled studies are produced. New therapies will then overcome problems due to insurance coverage, drug companies’ reluctance, and scientific guidelines. 

### 5.4. Surgery

Surgical correction of the urinary malformation has been advocated in children with CAKUT after a first febrile UTI [89]. The removal of an obstruction, as in the cases of urethral valves, a pelvic ureteral joint, or ureteral obstruction, is mandatory. The surgical correction of VUR has been largely performed with several techniques. Two old studies randomized children with a VUR to antibiotic prophylaxis or surgical correction; both studies demonstrated comparable outcomes [90,91]. A recent meta-analysis of randomized controlled trials in children with a VUR showed that surgical treatment may decrease UTI recurrence, while endoscopic treatment and conservative treatment are inferior to antibiotic treatment [72].

## 6. Guidelines

Several guidelines (GLs) on UTIs have been produced and periodically updated and have been recently reviewed [92]. Most GLs are limited to febrile UTIs in children less than 36 months old [38,39,40,41,42,43,44,45,46,54,56,88,93,94]. The major goals of these guidelines have been to make a prompt diagnosis and establish proper treatment, propose imaging studies in those who may have urinary tract malformation, and suggest medical or surgical treatment to avoid the progression of kidney damage. A discrepancy in guidelines on UTIs concerns methods in urine sampling, the definition of bacteriuria, prioritizing imaging procedures, the choice of antibiotics, length of treatment, and indication of prophylaxis. There is a need for more practical guidelines on UTIs for pediatricians and general practitioners who do not have access to hospital facilities. The Working Group “CAKUT, UTI and Bladder dysfunction” of the European Society of Pediatric Nephrology is working on a consensus.

## 7. Future Directions

Although UTIs are among the most common infections in children and a large number of scientific papers are continuously published, more studies are needed. Further knowledge of the mechanism involved in the aggression of bacteria to the urinary tract and kidney may stimulate new therapies. Bladder dysfunctions have a critical role and should be properly investigated. The role of urobiome and other factors involved in the natural defense against virulent bacteria deserve further investigation. It is crucial to understand the pathogenesis of scars in the kidney and detect risk factors leading to permanent kidney damage and consequent high blood pressure, eclampsia, and the possible loss of renal function. Permanent kidney damage is rarely observed in adults after a UTI, indicating a possible congenital predisposition. Future studies should concentrate on the genetic predisposition of kidney damage in children with UTIs. Urine cultures are still needed for the diagnosis of a UTI, but urine collection should be collected in a simpler way in small children, and quicker results should be offered; a quick chemical analysis of urine is possible with new technologies, which should be tested in children. Imaging studies should be limited to those giving crucial information for treatment, and more bladder function investigations should be carried out since bladder and bowel dysfunctions are associated with UTIs. Antibiotic treatment is mandatory in febrile UTIs, and several drugs are available, while chemoprophylaxis should be restricted to a minority of patients who are at risk of recurrent UTIs or kidney damage; however, the use of immunomodulators and probiotics may be an alternative. Surgery has been largely performed in the past; its role is now limited to very few cases. Several guidelines have been produced and updated, but we recommend preparing simple guidelines for general practitioners and pediatricians with limited resources.

## Figures and Tables

**Table 1 biomedicines-12-02582-t001:** Urine collection.

Urine Collection Methods	Advantages	Limitations
Urine Bag	Easy. Useful for dipsticks.	High risk of contamination.
Clean-Voided Urine	First choice of non-invasive methods. Reduced risk of contamination.	Time-consuming.
Catheter	First choice in hospital settings. Low contamination.	Invasive. Requires expertise.
Supra Pubic Aspiration	Gold standard. Very low contamination.	Invasive. Requires expertise.

**Table 2 biomedicines-12-02582-t002:** Cut-off values for a significant colony count in urine culture in national guidelines.

Method of Urine Collection	UK [38]	USA [39]	Canada [40]	Italy [41]	Spain [42]	Sweden [43]	Switzerland [44]	Asia [45]	India [46]
Urine Bag				>100,000 CFU/mL					
Clean-Voided Urine	Not defined		>100,000 CFU/mL	>50,000 CFU/mL	>100,000 CFU/mL	Not defined	>50,000 CFU/mL	>50,000 CFU/mL	10,000–100,000 CFU/mL
Catheter	Not defined	>50,000 CFU/mL	>50,000 CFU/mL	>10,000 CFU/mL	>10,000 CFU/ml	Not defined	>10,000 CFU/mL	>10,000 CFU/mL	>10,000 CFU/mL
Sovrapubic aspiration	Not defined	>50,000 CFU/mL	All growth	>10,000 CFU/mL	All growth	Not defined	All growth	All growth	All growth

**Table 3 biomedicines-12-02582-t003:** Imaging of UTIs in national guidelines.

	UK [38]	USA [39]	Canada [40]	Italy [41]	Spain [42]	Sweden [43]	Switzerland [44]	Asia [45]	India [46]
RUS	<6 months: All children >6 months Atypical o RUTI	All children	<2 years All children	All children	<6 months: All children >6 months Atypical o RUTI	<2 years All children >2 years Febrile or RUTI	All children with Febrile UTI	All children with Febrile UTI	All children
VCUG	<6 months Atypical or RUTI >6 months Atypical or RUTI + risk factor	Abnormal RUS If RUTI	<2 years Abnormal RUS RUTI	Abnormal RUS If RUTI If non-*E. coli* UTI		<2 years If abnormal RUS or DMSA <2 years If abnormal US and DMSA	Abnormal RUS RUTI febrile Risk factors	Abnormal RUS or DMSA RUTI	<2 years Abnormal RUS In febrile UTI If non-*E. coli* RUTI in
DMSA	<3 years Atypical or RUTI >3 years if RUTI	Not recommended	Acute scan	If VUR grade 4–5	Atypical or recurrent Acute scan	<2 years If abnormal RUS or RUTI <2 years If abnormal US and DMSA	Case by case	Acute scan Severe PN Hypodisplasia	If VUR grade 4–5 RUTI

RUS: renal ultrasound. VCUG: voiding cistography. DMSA: radionucleotide dimercaptosuccinic acid. RUTI: recurrent urinary tract infection.

**Table 4 biomedicines-12-02582-t004:** Treatment of febrile UTIs in national guidelines.

	UK [38]	USA [39]	Canada [40]	Italy [41]	Spain [42]	Sweden [43]	Switzerland [44]	Asia [45]	India [46]
Route of Administration	<3 months Ev >3 months oral	Ev: if unable to retain oral fluid Oral: otherwise	Ev: if unable to retain oral fluid Oral: otherwise	<3 months Ev or unable to retain fluid >3 months oral	<3 months Ev or unable to retain fluid >3 months oral	Ev: if unable to retain oral fluid Oral: otherwise	<2 months Ev or unable to retain fluid Oral: otherwise	<3 months Ev or unable to retain fluid >3 months oral	<2 months Ev or unable to retain fluid Oral: otherwise
Length of Treatment (Days)	10	7–14	7–10	10	7–10	10	7–10	7–14	7–10
Prophylactic antibiotics	NO	NO	VUR Grade 4–5	VUR Grade 4–5 >3 Febrile UTI	Obstructive Uropathy	VUR Grade 3–4–5 RUTI	VUR Grade 4–5 CAKUT Bladder dysfunction	VUR Grade 3–4–5 RUTI Obstructive Uropathy	

VUR: vesico-ureteral reflux. RUTI: recurrent urinary tract infection.

## Data Availability

The data are contained within the article.

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
