# Peer review of "Urinary Tract Infection in Children: An Up-To-Date Study"

_biomedicines, 2024, doi:10.3390/biomedicines12112582_

Round 1

Reviewer 1 Report

Comments and Suggestions for Authors

The manuscript presents an interesting critical review of current knowledge of bacterial UTIs in children, highlighting questions that have not yet been resolved. The review is concise and understandable; However, I have some observations:

-The title should be more specific, since they only address bacterial infections, I suggest: “

Urinary Tract Infection in Children: An UpToDate”

-The objective of the review is missing from the abstract.

- Authors should mention the limitations of the study or review. A limitation is that a systematic review was not performed.

-Delete the word Title in the manuscript title

-Authors must complement the following sections: Author Contributions; Funding; Institutional Review Board Statement; Informed Consent Statement; Data Availability Statement; Conflicts of Interest

-Authors should review the format of references and adhere to the guidelines of Biomedicines

Reviewer 2 Report

Comments and Suggestions for Authors

The article provides a comprehensive overview of urinary tract infections (UTIs) in children, covering various aspects such as epidemiology, diagnosis, treatment, and future directions. Minor issues should be addressed before acceptance is considered. 

1. The authors need to double check the title: replace title with the 

2. While the paper summarizes existing knowledge well, it could provide a deeper analysis of recent advances, particularly regarding emerging diagnostic techniques (e.g., the electronic nose) and novel treatment options (e.g., immunomodulatory therapies). Highlighting gaps in current research and the potential impact of these innovations could strengthen the paper.

3. The discussion on diagnostic methods is thorough but could be clearer with a tabular summary of the advantages and disadvantages of each method (e.g., bladder catheterization vs. suprapubic aspiration). Additionally, consider discussing newer, less invasive diagnostic techniques and their potential for broader adoption.

4. Diagnosis: The discussion on diagnostic methods is thorough but could be clearer with a tabular summary of the advantages and disadvantages of each method (e.g., bladder catheterization vs. suprapubic aspiration). Additionally, consider discussing newer, less invasive diagnostic techniques and their potential for broader adoption.

5. Treatment: The treatment section would benefit from a more in-depth discussion of alternative therapies, such as probiotics, cranberry products, and immunomodulation, and the supporting evidence for their use in pediatric patients. Clarifying which treatments are recommended based on the severity and recurrence of UTIs would also be useful.

6. Future Directions: This section is a bit vague. It could be strengthened by providing specific areas where further research is needed, such as the genetic predisposition to renal scarring or the development of rapid, non-invasive diagnostic tools.

7. What are the key challenges in implementing non-antibiotic treatments (e.g., probiotics, D-mannose) in clinical practice? Discussing barriers to adoption, such as regulatory issues or lack of robust evidence, would provide a more comprehensive view.

Reviewer 3 Report

Comments and Suggestions for Authors

Authors are advised to add some figures and tables and work on the presentation of the review article. The topic is very interesting but major work is needed to improve the presentation and organization of the review article. I am feeling sad to say it, but in its current form I would not recommend the publication of this review article.  

The review article covers a very interesting topic of bacterial infection of urinary track in children.   Although the topic is very interesting but the review is lacking vigor and does not address relevant gaps in the filed that can improve the already available knowledge about the topic.  I have recommended the rejection of the articles as there are major concerns about the organization and presentation some specific improvements such as consistency, coherence and presentation. The article has no figures and tables that are crucial for making the complex point clear.   The conclusion section is completely missing, neither the abstract nor any other part of the review clarify the main objective of the article. The authors contribution statement is not addressed.  After noticing all theses concerns, I recommend the rejection of the manuscript.

Round 2

Reviewer 3 Report

Comments and Suggestions for Authors

Dear Authors I believe the review article you have submitted covers a very interesting topic of bacterial infection of urinary track in children. Urinary tract infections (UTIs) are the second most common bacterial infections in children. The authors claimed that several guidelines on UTIs are available but a simple one for general practitioners is needed. Its a very intresting topic but I am not sattisfied with the way information has been presented and I will rewrite my opinion that the article lacks vigor to be approved for publications.  Although the topic is very interesting but the review  does not address relevant gaps in the filed that can improve the already available knowledge about the topic.  The article has no figures and tables added by the authors are not readable and some columns are lost.  Tabels are not introduced and cited in the text (table 2, table 3 and table 4).  The conclusion section is completely missing as well in the revised manuscript.  After noticing all theses concerns, I  once again recommend the rejection of the manuscript. The authors are unwilling or unable to improve the manuscript.

Comments on the Quality of English Language

The english language is missing the cosistency and this reviewer has noticed mistakes such as " Since symptoms of UTIs are less evident in newborn, infants and small children early diagnostic tools, including  invasive imagine procedures, have been advocated but are not necessary in most cases".

"A permanent kidney damage is rarely observed in adults after an UTI indicating a 334 possible congenital predisposition"

Author Response

Please see the attachment and the revised manuscript
